# Inter-Floor Noise Monitoring System for Multi-Dwelling Houses Using Smartphones

**Suhyun Kang [1], Seungho Kim [2], Dongeun Lee [3] and Sangyong Kim [1,\*]**

[1]   School of Architecture, Yeungnam University, 280 Daehak-ro, Gyeongsan-si, Gyeongbuk 38541, Korea; yp043422@ynu.ac.kr

[2]   Department of Architecture, Yeungnam University College, 170 Hyeonchung-ro, Nam-gu, Daegu 42415, Korea; kimseungho@ync.ac.kr

[3]   School of Architecture & Civil and Architectural Engineering, Kyungpook National University, 80 Daehak-ro, Buk-gu, Daegu 41566, Korea; dolee@knu.ac.kr

\*   Correspondence: sangyong@yu.ac.kr; Tel.: +82-53-810-2425

**Abstract:** The noise between the floors in apartment buildings is becoming a social problem, and the number of disputes related to it are increasing every year. However, laypersons will find it difficult to use the sound level meters because they are expensive, delicate, bulky, etc. Therefore, this study proposes a system to monitor the noise between the floors, that will measure the sound and estimate the location of the noise using the sensors and applications in smartphones. To evaluate how this system can be used effectively within an apartment building, a case study has been performed to verify its validity. The result shows that the mean absolute error (MAE) between the actual noise generating position and the estimated noise source location was measured at 2.8 m, with a minimum error of 1.2 m and a maximum error of 4.3 m. This means that smartphones, in the future, can be used as low-cost monitoring and evaluation devices to measure the noise between the floors in apartment buildings.

**Keywords:** inter-floor noise; multi-dwelling houses; smartphone application; real-time monitoring system

## 1. Introduction

### 1.1. Background

Population concentration due to urbanization has led to housing shortages, and many cities opted for the construction of multi-dwelling houses, which can be supplied in large quantities at a relatively low cost, as a solution [1]. In multi-dwelling houses, however, the residents are easily exposed to the noises of neighbors, as the walls and slabs are shared with other households. The continuous exposure to external noises of the residents of multi-dwelling houses may cause physical and mental health problems, such as high blood pressure, annoyance, and sleep disorders [2–4]. As such, inter-floor noise has also caused discord amongst neighbors, including an elevated number of disputes, assaults, and even arson [5–7].

To address disputes related to inter-floor noise, it is essential to secure objective noise data. Sound level meters are generally used to obtain objective noise data. It is difficult, however, for non-experts to use sound level meters, because they are expensive, delicate, and bulky [8]. The recent technical development of smartphones has opened up a possibility where they can be used as substitutes for sound level meters [9–11].

Smartphones are powerful mini-computers with various sensors (e.g., microphones, accelerometers, gyroscopes, and GPS) and are owned by the majority of the population. They can be used as low-cost noise monitoring tools with available broadband internet access [12].

A number of studies have been conducted lately to examine the accuracy of smartphone noise measurement applications (apps). Murphy and King [11] tested the accuracy of several noise measurement apps on two platforms (Android and iOS) using 100 smartphones. The test results showed that one of the apps was very accurate in measuring the noise levels with errors less than ±1 dB from the actual sound levels in the reference value range. The conducted study indicated that noise measurement apps have a potential to be used as sound level meters in the future. Zamora et al. [13] proposed environmental noise-sensing units using smartphones. According to these experimental results, if the smartphone application is well tuned, it is possible to measure noise levels with an accuracy degree comparable to professional devices for the entire dynamic range typically supported by microphones embedded in smartphones. Garg et al. [8] proposed an averaging method for accurately calibrating the noise acquired through a smartphone microphone. This method achieves an accuracy of 0.7 dB.

Smartphones also provide an inexpensive and flexible infrastructure for the measurement of overall environmental noise (e.g., noise and air pollution) in cities. Various related studies have shown that smartphone apps are useful for environmental monitoring evaluation [14–17]. Although the aforementioned studies verified the accuracy of smartphone noise measurement apps and their potential as environmental monitoring tools, studies on the possibility of using smartphones to address the inter-floor noise problem are not sufficient.

The problem to be solved in relation to inter-floor noise is to identify the noise types and locations of those noise sources [18]. This is important, since some disputes have resulted from misunderstanding of the noise sources by listeners [18]. Most studies on inter-floor noises, however, are focused on noise measurement [3,19], noise reduction measures [20,21], and annoyance measurement [22,23]. If smartphones can identify objectively and reliably the noise source locations and noise types in real time, they can contribute to dispute mediation.

## 1.2. Motivation and Objective

Inter-floor noise is transmitted to neighboring households in multi-dwelling houses, and unpleasant sounds disturb other house residents. In South Korea, where most people live in multi-dwelling houses, 88% of the apartment residents are under stress due to inter-floor noise [24]. In South Korea, most apartments have been constructed in the wall column structure style since the 1980s, due to reasons of constructability, economic efficiency, and a reduction in the construction period. In apartments with the wall column structure, all four apartment sides are made of concrete, with a large vibration transfer coefficient. Thus, the airborne sound that is generated on the upper floor and the vibration that is generated at the bottom of the upper floor are easily transferred to the lower floor [25].

In particular, the wall column structure apartments built before 2005 in Korea generally used a concrete slab thickness ranging from 135 mm to 150 mm, but in recent years, with the emergence of frequent inter-floor noise problems, a new regulation was established to standardize the slab thickness to be at least 210 mm [3]. Despite the legal regulations on the slab thickness, the number of complaints related to inter-floor noise has increased from 8795 in 2012 to 28,231 in 2018 (Figure 1).

This phenomenon appears to have occurred because there was no solution for noise mitigation for the existing apartments built before 2005, when the regulations on the slab thickness were enacted. The regulations can be applied only to the newly built apartments because improved construction methods, such as reinforced thicknesses of the walls and floor slabs and application of floating floors, have not been made available for the existing apartments. However, there has been an increase in the number of complaints related to inter-floor noise in new apartments built under new regulations. The study conducted by Park, Lee and Lee [3] verified that the slab thickness did not have any effect in lowering the indoor noise level.

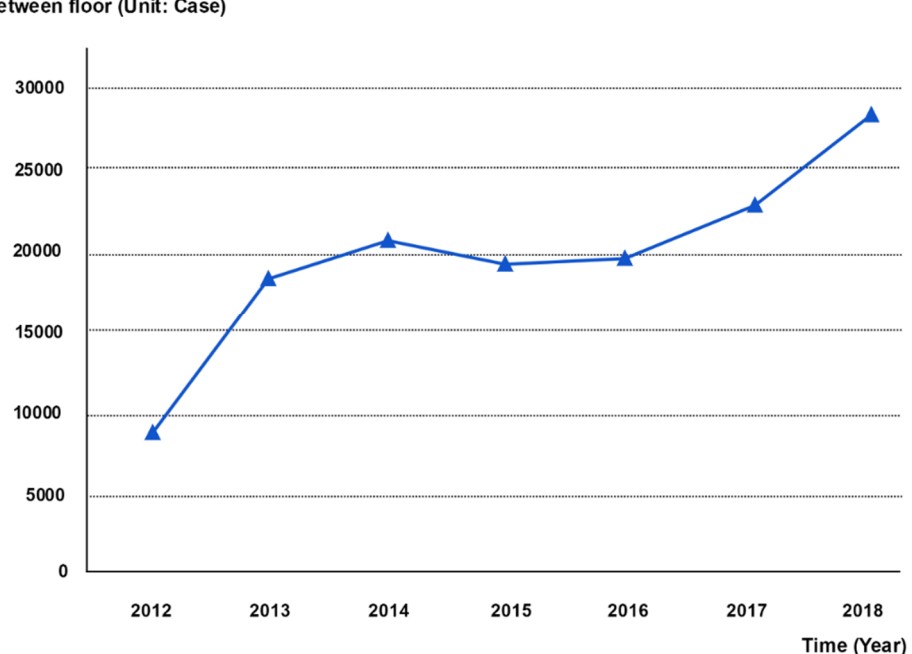

**Figure 1.** Trend of inter-floor noise complaints (Korea Environment Corporation).

The increase in the inter-floor noise complaints has led to conflicts and disputes among neighbors [26]. Emotional reactions to noise problems even led to a number of retaliatory crimes between neighbors, such as arson and murder [27]. As the conflicts caused by inter-floor noise expanded to a social problem, the South Korean government established a 'center for inter-floor noise mitigation between neighbors' in 2012, to oversee the disputes related to inter-floor noise. The center, however, has no legal rights and on-site investigation for objective noise measurement and shows some limitations in solving the inter-floor noise problem, due to a lack of manpower. The inter-floor noise problem is still unsolved, and thus, more effective measures are required to resolve the occurring disputes.

As noise is judged from a subjective perspective due to its environmental nature, conflicts due to a difference in opinions cannot be avoided. To resolve such conflicts, it is necessary to prove the fact that a noise level higher than the inter-floor noise criterion occurred, state its duration, and the degree of damage caused. Therefore, this study proposes an inter-floor noise monitoring system for measuring the inter-floor noise and estimating the noise time and location, by utilizing sensors and mobile applications of widely available smartphones. The proposed system enables recording various data related to inter-floor noise, and it is expected to be used as an important tool for resolving disputes related to inter-floor noise in the future.

## 2. Research Method

In this study, a system to monitor inter-floor noise using smartphones is proposed. To verify the validity of the system, apartment B, completed in 1996 and located in Gyeongsan City, Gyeongsangbuk-do, South Korea, was selected as a case study site. For inter-floor noise monitoring, an inter-floor noise monitoring application was developed using sensors built into smartphones. To this end, the functions of such sensors were identified and used to achieve the target functions for the inter-floor noise monitoring system.

Table 1 shows the smartphone sensors and their functions, that were used in this study in order to implement the developed application. The microphone was used to obtain the sound pressure level (SPL). The accelerometer and gyroscope were used to measure the vibration acceleration level (VAL) created by a heavy impact on part of a building. Moreover, GPS was used to locate the smartphone

and to measure the timing of the occurring noise. Wi-Fi was used to transfer the obtained inter-floor noise information to a server.

**Table 1.** Smartphone sensor features and their utilization for the application.

| Sensor Type | Description | Application |
|---|---|---|
| Microphone | Detects sound signals and converts them into an electrical signal | Sound detecting and record/Sound Pressure Level (SPL) measurement |
| Accelerometer | Measures the acceleration force in m/s$^2$ on all three physical axes ($x$, $y$, $z$) | Distinguish between air-borne sound and floor impact sound/Vibration Acceleration Level (VAL) measurement |
| Gyroscope | Measures a device's rate of rotation in rad/s on all three physical axes ($x$, $y$, $z$) | |
| GPS | Positioning and provides time information | Identify the location of the noise measuring device and the time of noise occurrence |
| Wi-Fi | Wireless networking | Noise Data Transmission |

The developed inter-floor noise monitoring application requires a certain level of sound as a baseline for determining inter-floor noise. In this study, the legal criteria existing for the case study site (i.e., for South Korea) were applied. Inter-floor noise is largely divided into floor impact noise (e.g., running and walking sounds), which is generated when the energy is applied directly to the floor, and airborne sound (e.g., conversation and musical instrument sounds). Therefore, when a floor impact occurs, inter-floor noise must be determined by measuring the SPL of the lower floor and the vibration acceleration level generated by construction components (e.g., ceilings, walls, and windows). Table 2 shows the criteria for each type of inter-floor noise, as specified by the Ministry of Environment and the Ministry of Land, Infrastructure and Transport of South Korea.

**Table 2.** Criteria of noises between floors (Korea Ministry of Government Legislation).

| Classification | | Standard Value (Unit: dB) | |
|---|---|---|---|
| | | Day Period | Night Period |
| Floor impact sound | A minute equivalent sound level (LAeq 1 min) | 43 | 38 |
| | The highest sound level (LAmax) | 57 | 52 |
| Air-borne sound | A five minute equivalent sound level (LAeq 5 min) | 45 | 40 |

In the case of floor impact noises, inter-floor noise is determined when 'LAeq 1 min' exceeds 43 dB in the daytime and 38 dB at night, or when 'LAmax' exceeds 57 dB in the daytime and 52 dB at night. LAeq 1 min corresponds to the average value of noise measured for one minute, using a sound level meter. LAmax denotes noise with the highest dB value among the noises generated during the measurement period. In the case of airborne sounds, inter-floor noise is determined when 'LAeq 5 min' exceeds 45 dB in the daytime and 40 dB at night. The length of airborne noise detection was extended to five minutes, to reflect the long-lasting characteristics of television noise or musical instrument sounds. Therefore, in this study, inter-floor noise was determined by applying the above-mentioned criteria to the smartphone application.

## 3. Construction of the Monitoring System for Measurement of Inter-Floor Noise and Estimation of Noise Source Location

### 3.1. System Design

Figure 2 shows the configuration of the proposed monitoring system for the measurement of inter-floor noise levels and the estimation of noise source locations. In general, the system contains four steps. In the first step (the inter-floor noise sensing step), noise and vibration data are obtained

from the place where data acquisition is required. Data is collected using the microphone, gyroscope, and accelerometer embedded in a smartphone. The decibel value and vibration velocity (i.e., noise data) are acquired every second, and the surrounding noise is recorded every minute. The acquired noise and vibration data are then transferred to a web server through Wi-Fi wireless communication in the second step (the inter-floor noise data transfer step). In this case, the transferred data consist of the ID and location of the measuring device, noise acquisition time, decibel level (dB) values, and vibration velocity ($m/s^2$). The web server stores the transferred data in a database in real time.

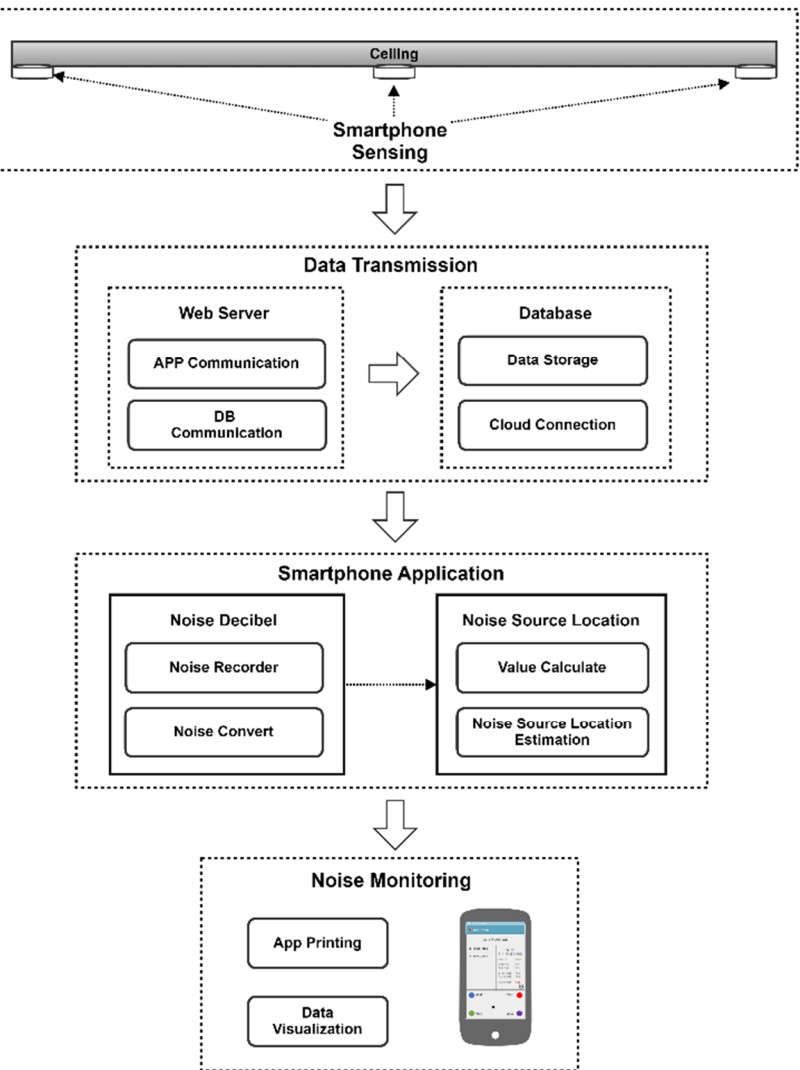

**Figure 2.** System Architecture.

Figure 3 shows a schema of tables that are stored in a database. The database consists of a number of tables, such as the NoiseHistory, DeviceList, and RecordList. Each table contains noise data, information on noise measuring devices, and recorded files. In the NoiseHistory table, the ID of the device that transferred the data, acquisition time, decibel values, and vibration velocity are stored. When the decibel value is higher than the threshold, "1" is recorded in the noise field. In this instance, noise is determined using the criteria displayed in Table 1. Information on the ID and location of each device is stored in the DeviceList table. Information on the files recorded by each device is stored in the RecordList table. In the third step, the developed application estimates the location of the noise source, based on the records stored in the database. The application stores the noise data values in real time, converts them into decibel values, and determines the noise location using the

estimation algorithm. In the final step, the acquired inter-floor noise information is visualized on the user's smartphone screen.

| NoiseHistory, Major key : ID + date | | |
|---|---|---|
| Field name | Data Type | Explanation |
| ID | int | Measurement device ID |
| date | datetime | Data acquisition time (yyyyMMddHHmmss) |
| dB | float | Decibel |
| Speed | float | Vibration acceleration(m/s^2) |
| noise | int | In regards of noise (Y=1, N=0) |

| DeviceList, Major key : ID | | |
|---|---|---|
| Field name | Data Type | Explanation |
| ID | int | Measurement device ID |
| loc_x | int | x-axes with device installed |
| loc_y | int | y-axes with device installed |

| RecordList, Major key : ID | | |
|---|---|---|
| Field name | Data Type | Explanation |
| ID | int | Measurement device ID |
| date | datetime | Recording time(yyyyMMddHHmm) |
| path | varchar | File storage path |

**Figure 3.** Database Schema.

Figure 4 shows the application execution screen. The information that can be found in the application includes the timing of occurring noise, the noise measurements at that time, the estimated noise location and the noise type. The location at which the noise occurred is displayed on the floor plan of the measurement site and is located at the bottom of the application. The noise type (e.g., floor impact or airborne noise) can be determined using the recorded vibration values. It is determined as floor impact noise if there is vibration information when the noise occurred, or as airborne noise if there is no vibration information available.

### 3.2. Noise Source Location Estimation Method Used in This Study

Previous studies on sound source location estimation have been conducted using specialized equipment, such as microphone arrays. Those studies were also arranged for limited experimental environments [28,29]. The proposed system, however, uses only smartphones, thereby providing a method for many people to easily estimate noise source locations. In this study, an attempt was made to estimate noise source locations using differences in the sound intensity. For this method, hardware configuration and operation are very simple, even though it is difficult to calculate the exact distance to the sound source. The purpose of this study is not in finding the exact location of noise, but rather in estimating the approximate noise source occurrence area.

Due to the nature of sound, a lower decibel value is measured as the distance increases. Based on this phenomenon, a method of estimating noise sources using the proportions of the decibel values measured through four smartphones is described. As shown in Figure 5a, it is assumed that noise measurement devices ($T = \{T_1, T_2, T_3, T_4\}$) are placed in the form of a grid in two-dimensional coordinates. Each device has a decibel value (dB) and coordinate information ($x, y$). In this study, among the noise measuring devices ($T$), three devices ($S_1, S_2, S_3$) are arbitrarily selected according to the decibel level to locate the noise source. As shown by Equation (1), among the devices ($T$), the device with the largest decibel value (dB) is designated as $S_1$.

$$S_1 = \text{Max.db}(T) \tag{1}$$

For example, when a noise or vibration takes place, assuming that the highest decibel value was observed in $T_1$ among the devices ($T$), the $T_1$ device is set as $S_1$. Subsequently, as shown by Equation (2), the device ($T$) located on the horizontal line of $S_1$ is selected as $S_2$.

$$S_2 = \begin{cases} \textit{if } S_1 \cdot y = T_2 \cdot y \textit{ then } T_2 \\ \quad\quad \textit{else } T_3 \end{cases} \tag{2}$$

Here, $S_2$ is a device which has the same $y$-coordinate value as, but a different $x$-coordinate value to, $S_1$. Lastly, as expressed by Equation (3), the device having the largest decibel value among the devices other than the devices designated as $S_1$ and $S_2$ is selected as $S_3$.

$$S_3 = \{t \cdot y = S_1 \cdot y \wedge t \cdot x \neq S_1 \cdot x \mid t \in T\} \tag{3}$$

When it is assumed that $T_1 \cdot \text{db} = 80$, $T_2 \cdot \text{db} = 40$, and $T_3 \cdot \text{db} = 60$, the placement of $S_1$, $S_2$, and $S_3$ can be expressed as shown in Figure 5b. In this case, the approximate values of X and Y that serve as the estimated location coordinates of the noise source are obtained using Equations (4) and (5).

$$X = \begin{cases} \textit{if } S_1 \cdot x > S_2 \cdot x \textit{ then } \frac{S_1 \cdot db}{S_1 \cdot db + S_2 \cdot db} \cdot \textit{width} \\ \quad\quad \textit{else } \frac{S_2 \cdot db}{S_1 \cdot db + S_2 \cdot db} \cdot \textit{width} \end{cases} \tag{4}$$

$$Y = \begin{cases} \textit{if } S_1 \cdot y > S_3 \cdot y \textit{ then } \frac{S_1 \cdot db}{S_1 \cdot db + S_3 \cdot db} \cdot \textit{height} \\ \quad\quad \textit{else } \frac{S_3 \cdot db}{S_1 \cdot db + S_3 \cdot db} \cdot \textit{height} \end{cases} \tag{5}$$

Width means the distance between $S_1$ and $S_2$, and height is calculated as the distance between $S_1$ and $S_3$. Figure 5 shows the estimated noise source locations using Equations (4) and (5).

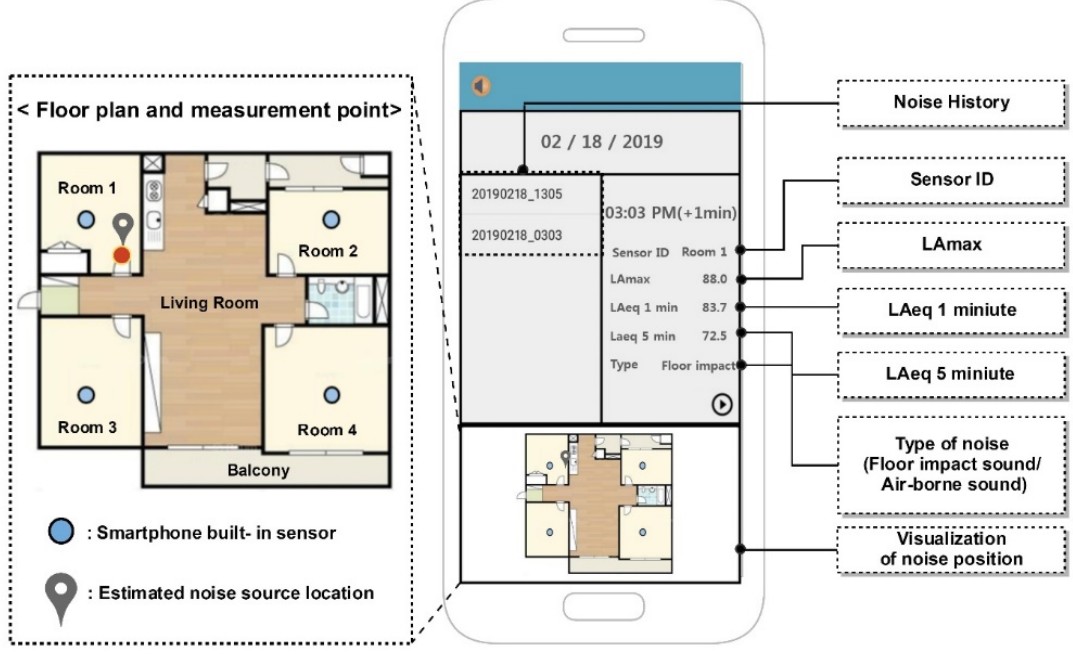

**Figure 4.** Application execution screen.

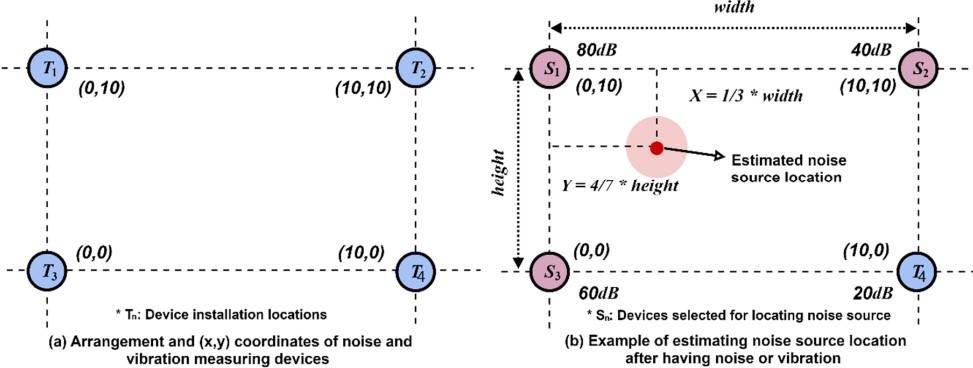

**Figure 5.** Methods for estimating the noise source location.

## 4. Experiment and Performance Evaluation

### 4.1. Experiment Overview

In this study, inter-floor noise data were acquired using four smartphones to estimate the noise source locations, and one smartphone was used to display the inter-floor noise data in real time for the user. Thus, a total of five smartphones were used in the experiment.

#### 4.1.1. Software/Hardware Configuration

Table 3 shows the software components used in the experiment. In this study, JSP programming language was used based on Apache Tomcat (a web application server—WAS) in a Windows 10 Pro operating system for system development. Moreover, the database was managed by linking Apache Tomcat with MySQL. Android 5.0 APIs was used as an operating system to control smartphones.

**Table 3.** Software component.

| Component | Explanation |
|---|---|
| Computer OS | Windows 10 pro |
| Web Application Server (WAS) | Apache Tomcat |
| Programming Language | JavaServer Pages(JSP) |
| Database management system (DBMS) | MySQL |
| Mobile OS | Android 5.0 APIs |

Table 4 shows the hardware components used in the experiment. As the noise source locations were estimated using the differences in the sound intensity acquired from four measuring devices, only one smartphone model was used for the same conditions. Hardware was easily obtained, and devices with the sensors required for system implementation were selected.

**Table 4.** Hardware component.

| Classification | Component | Specification |
|---|---|---|
| Built | Dimensions, Weight | 146.8 × 75.3 × 8.9 mm, 163 g |
| | Display Size | 13.3 cm (5.25 inches) |
| Platform | OS | Android |
| | AP | 4 Core, 1.2 Ghz |
| | CPU | Quad-core 1.2 GHz Cortex-A7 |
| | GPU | Adreno 305/400 MHz |
| | RAM | 1.5 GB/LPDDR2 SDRAM |
| Communications | Network | 4 G LTE |
| | WIFI | 802.11 b/g/n/ac, dual band |
| | GPS | ○ |
| Sensor | Microphone | ○ |
| | Accelerometer | ○ |
| | Gyroscope | ○ |

### 4.1.2. Experimental Environment and Method

To evaluate the performance and applicability of the proposed system to measure inter-floor noise and track the noise source locations, the experiment was performed in an apartment that serves as a representative for the residential type of multi-dwelling houses. Table 5 shows the overview of the experiment site.

**Table 5.** Profile of the experiment place.

| Type (P'yong) | 45 (148 m$^2$) | |
|---|---|---|
| Dimension [mm] | Lenght | 11,500 |
| | Width | 12,100 |
| | Ceiling height | 2200 |
| Area of measurement room [m$^2$] | Room 1 | 10.51 |
| | Room 2 | 13.06 |
| | Room 3 | 13.85 |
| | Room 4 | 5.62 |
| | Livingroom | 28.36 |
| Thickness of slab [mm] | 180 | |
| Measurement point | 4 point | |

The floor of the experiment site consisted of a reinforced concrete slab (180 mm), insulating materials (20 mm), lightweight concrete (40 mm), cement mortar (40 mm), and floor finishing materials (Figure 6). To collect noise and vibration data, smartphones were installed on the ceiling of each room (Figure 7). The exact installation locations can be found on the floor plan (Figure 4). The smartphone located at the bottom left corner was then designated as the origin, and the scales were marked at 24.2 cm intervals in the horizontal direction and at 23 cm intervals in the vertical direction.

As for the noise generation type, real impact sources (e.g., human footsteps and dropped objects) were used rather than standard impact sources (i.e., impact balls), to create an environment similar to real inter-floor noise in the experiment. At certain points over the ceiling, random noises were generated for over 20 s at a time (i.e., impacts of >70 dB, human voices, musical instrument sounds).

The experiment was repeated 100 times, whilst the noise occurrence locations were randomly changed, and the actual noise occurrence locations were then compared to the estimated locations displayed in the application.

### 4.2. Experimental Evaluation Method and Results

To evaluate the performance of the system, the errors between the actual noise occurrence locations and the estimated noise source locations were obtained using the mean absolute error (*MAE*). *MAE* was calculated using Equation (6).

$$\text{MAE} = \frac{1}{n} \sum_{i=1}^{n} distance(rpoint_i,\ ePoint_i) \tag{6}$$

where $rpoint_i$ is the epicenter of the *i*-th actual noise and $ePoint_i$ is the estimated location of the *i*-th noise. Figure 8 shows the *distance* function to obtain the absolute error between the actual noise epicenter and the estimated location.

Table 6 shows the experiment results. The calculated mean absolute error (MAE) was 2.8 m, while the minimum and maximum errors were 1.2 and 4.3 m, respectively.

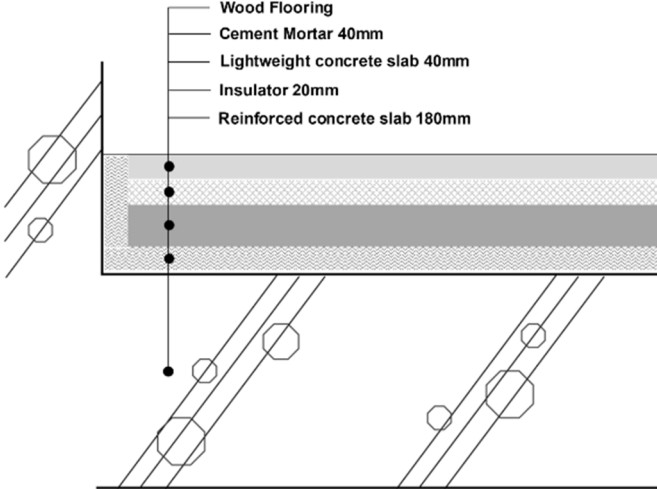

**Figure 6.** Cross section of the slab.

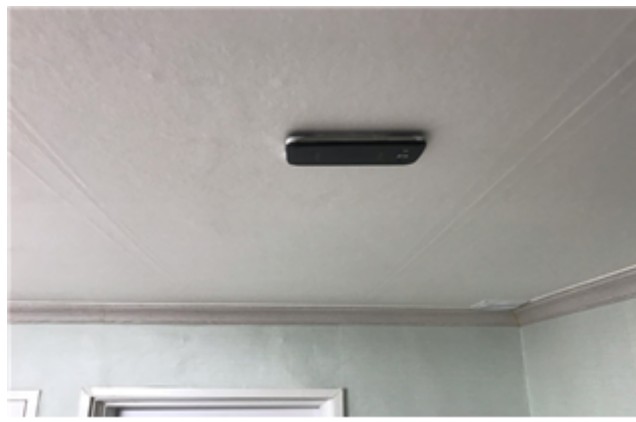

**Figure 7.** A smartphone installed on the ceiling of the multi-dwelling house.

```
function : distance(Point1, Point2)
  1   df_x = Point1.x - Point2.x
      // The difference in x-cordinates
  2   df_y = Point1.y - Point2.y
      // The difference in y-cordinates
  3   distance = root(df_x * df_x + df_y * df_y)
      //Distance between two points using Pythagoras
  4   return distance
```

**Figure 8.** Distance function.

Exact noise source locations could not be identified with the calculated values, but they were sufficient to distinguish among noise occurrence areas (Room 1, Room 2, Room 3, or Room 4) of the study site. Therefore, the proposed system performed the following four target functions using the smartphone sensors and the developed application: (1) it displayed the degree of inter-floor noise (dB) and recorded its values in the application by using the smartphone microphone devices; (2) it detected vibration using accelerometers and gyroscopes and classified the types of inter-floor noise (e.g., floor impact noise, airborne noise); (3) it estimated the noise source locations using the differences in the sound intensity and visualized the locations on the apartment floor plan; and (4) it provided reports of inter-floor noise on an hourly, daily, and monthly basis. Such reports are generated based on the information stored in the database, so that the recorded data can be accessed if a dispute occurs.

**Table 6.** Measured differences (m) between the actual and estimated noise sources.

|  | *rpoint$_i$* (x, y) | *ePoint$_i$* (x, y) | **Distance (*rpoint$_i$*, *ePoint$_i$*) (m)** |
|---|---|---|---|
| 1 | (24.2, 23.0) | (25.3, 24.4) | 1.8 |
| 2 | (1640.3, 5.1) | (1638.6, 4.4) | 1.9 |
| 4 | (765.6, 121.0) | (767.5, 123.0) | 2.8 |
| 7 | (538.5, 98.9) | (532.4, 96.1) | 4.3 |
| 10 | (364.1, 483.6) | (36.0, 482.7) | 1.2 |
| 14 | (1161.6, 1031.0) | (1164.4, 1031.6) | 2.9 |
| 18 | (219.5, 643.5) | (217.8, 641.0) | 3.1 |
| 20 | (721.8, 469.3) | (723.5, 472.7) | 3.8 |
| 24 | (836.2, 563.7) | (838.9, 565.5) | 3.3 |
| 35 | (1638.6, 4.4) | (1641.3, 7.5) | 3.8 |
| 42 | (121.3, 689.9) | (122.2, 690.2) | 1.4 |
| 50 | (766.7, 123.2) | (765.6, 121.3) | 2.2 |
| 65 | (25.6, 25.2) | (24.2, 23.4) | 2.3 |
| 70 | (839.5, 657.8) | (836.2, 655.8) | 3.9 |
| 75 | (387.0, 583.5) | (383.5, 581.4) | 4.0 |
| 80 | (217.8, 641.88) | (220.8, 648.2) | 4.1 |
| 85 | (741.4, 1021.4) | (740.0, 1019.1) | 2.7 |
| 90 | (482.2, 65.6) | (485.0, 67.5) | 3.4 |
| 95 | (38.8, 139.9) | (38.7, 193.1) | 1.3 |
| Mean |  |  | 2.8 m |

## 5. Conclusions

This study proposed a system capable of monitoring inter-floor noise in real time, using smartphone sensors and a developed application. The designed noise monitoring system makes it possible to record the timing of the noise and its type (i.e., floor impact noise or airborne noise), acquire the exact noise values (e.g., LAeq 1 min, LAmax, and LAeq 5 min), estimate the location where the noise took place, and keep record of noise files by using the smartphone application.

An experiment was performed to evaluate the performance of the system and its applicability to multi-dwelling houses. The experiment results showed that the mean absolute error (MAE) was 2.8 m, and the minimum and maximum errors constituted 1.2 and 4.3 m, respectively. Although the exact locations of the noise sources could not be identified with these values, it was possible to establish the noise occurrence areas by a room on the apartment floor plan. Therefore, it is concluded here that the tested system can easily acquire objective noise data without any help of agencies specializing in inter-floor noise measurements. It is also expected that this system can reduce unnecessary misunderstandings among neighboring residents, by estimating the types and locations of inter-floor noise. Accordingly, in the case of having an inter-floor noise dispute, the inter-floor noise data stored in the database can be accessed through the application.

While a recent increase in the number of discarded smartphones has caused problems such as the waste of resources and pollution of soil by heavy metals, recycling the discarded smartphones using the results of this study is expected to contribute to solving social problems. However, given that the proposed system does not have any noise-data filtering feature, there is a possibility of violating the privacy of others. Therefore, a number of criteria is yet to be met for the future application usage: (1) a method of estimating exact noise locations using smartphone sensors has be developed, (2) a calibration method to measure the accuracy of sound should be administered, and (3) the privacy of neighbors and personal data collection should be sufficiently protected.

**Author Contributions:** Conceptualization, S.K. (Suhyun Kang), S.K. (Sangyong Kim), S.K. (Seungho Kim), and D.L.; data curation, D.L.; formal analysis and investigation, S.K. (Suhyun Kang), S.K. (Sangyong Kim); methodology, S.K. (Suhyun Kang), S.K. (Sangyong Kim), S.K. (Seungho Kim); resources, D.L.; software, S.K. (Suhyun Kang) and S.K. (Seungho Kim); supervision, S.K. (Sangyong Kim) and D.L.; validation, S.K. (Suhyun Kang), S.K. (Sangyong Kim), S.K. (Seungho Kim), and D.L.; visualization, S.K. (Suhyun Kang); writing—original

draft, S.K. (Suhyun Kang), S.K. (Seungho Kim); writing—review and editing, S.K. (Sangyong Kim) and D.L. All authors have read and agreed to the published version of the manuscript.

**Funding:** This work was supported by the National Research Foundation of Korea (NRF) grant funded by the Korea government (MSIT) (No. NRF-2018R1A5A1025137).

**Conflicts of Interest:** The authors declare no conflict of interest.

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
