# Peer review of "Inter-Floor Noise Monitoring System for Multi-Dwelling Houses Using Smartphones"

_sustainability, doi:10.3390/su12125065_

Round 1

Reviewer 1 Report

Brief summary – The article aims to propose a system for the monitoring of the noise between floors in multi-storey residential buildings, able to measure the sound level and estimate the location of the noise in real time using the sensors and applications in smartphones as low-cost and handy alternative to traditional sound level meters.

Broad comments - The general structure of the paper is correctly organized and articulated, the different sections are balanced as for level of details, the Bibliographic reference section is rather essential.

The English language is fluent and understandable with just few imprecisions in the sentence structuring.

The topic is in itself very interesting and a practical one. It meets a specific problem in its own reference field, since the proposed solution allows to overcome a widespread problem, such as the availability of new easy-to-use and low-cost tools for measuring sound levels and solving legal disputes between neighbours.

The research design is consistent and scientifically correct, although not really innovative. The proposed research method is reliable and, in general, well described, as are data used. The research is sufficiently detailed in order to be reproduced, and it correctly identifies and acknowledges its inherent limitations. The results are consistent with the objective of the study. As a whole, it is quite simple and not cutting-edge, but it answers to a real and practical need in buildings’ use.

Although it is consistent with the general theme of building technologies, it is to some extent out of the Special Issue's stated focus, i.e. the sustainability of buildings and infrastructure, which most impacts climate change, by minimizing energy and resource use and waste reduction throughout buildings’ life cycle. None of these issues is tackled in the paper, so it is not really consistent with the Special Issue.

As for the experimentation carried out, one specific point is the choice to use real sources (human footsteps and dropped objects) rather than standard sources. Although the authors explicitly mention the aim to reproduce a close-to-real environment, the decision is not congruent and consistent with the type of investigation. Since the goal of the experiment is to deduce statistical data such as LAmax, LAeq and MAE through a repetition of many equal events, identical conditions should be assured. That is, vary the source position and type of source all other conditions being equal. This is not the case with real, non-standard events, such as human stepping. The use of real sound sources can undermine the reliability of calculations based on comparison, as, differently from standardized ones, they cannot assure a perfectly identical reproduction. So it is not fully understandable why standardized sources were not used, given that the related equipment is quite easy to arrange and that the aim of the work is to check the reliability of devices.

Finally, the relation between the proposed system and resource waste/recycling (lines 295 – 298) is not clear, though it could be interesting. This part should be deepened in order to make the work closer to the Issue topic.

It is overall an acceptable work, but my warm suggestion is to propose the article for publication in a more appropriate Special Issue of the same Journal, for which it could be more relevant.

Specific comments

Figure 1 is not consistent with the text in Lines 78-80; furthermore, paragraphs 1.1 and 1.2 have, for a large part, very similar contents, and should be differentiated.

Lines 83-84: “as per a new regulation was strengthened to over 210 mm ass the inter-floor noise problem emerged”: unclear sentence, please check grammar and possibly rephrase

Line 118 “and to measurer the timing”: please check grammar

Table 2 - LAmax and LAeq 1minute values: please check (swap)

Lines 188–209: the whole formulation is not clear: please explain what is indicated with S1 • x and S2 •y

Lines 191–193: the whole period is obscure, please rephrase the sentence

Table 3 – “Wed Application Server”: please, check

Lines 291-293: actually, a MAE value of 3.4m is largely compatible with the linear dimensions of rooms, then the proposed solution might not be able to solve disputes between the affected owner and two upper adjacent apartments as for the exact position of the sound source and the correct attribution of responsibilities. This point should be considered among the study’s limitations.

Reviewer 2 Report

The introduction and the literature review should talk about the use of smartphone sound measurement apps in a broader way. What has been the impact of smartphones in the areas of noise research and noise control in other areas and not only talking about noise complaints in apartments. For example, many scholars have discussed the implications of these apps in the workplace environment (see Williams W. and Sukara Z. [2013]. Simplified noise labeling for plant or equipment used in workplaces. Journal of Health and Safety, Research and Practice, Vol. 5 (2), 18-22.) Older adults who have trouble hearing could use this app for when they are welcoming grandchildren at home and they want to turn on the T.V. for example. If the lit review was broader I think that it would capture a broader audience and also show what are the implications of the technology.

Can the authors elaborate on the South Korean government noise regulations? Can you also say a little bit about how regulations come into being and who enforces them? For example, in the U.S. is usually the health department and local law enforcement agencies like zoning officers or the local police to enforce the noise regulations. Usually noise regulation is not about the materials of the home being build but more about prohibiting loud noise at night between, for example, hours of 10:00 p.m. and 7:00 a.m. In addition, noise regulation tends to sets limits for extremely loud noise during daytime hours. Can you define the noises that are (un)reasonable, what is someone is remodeling the place or vacuuming?  

Are noise violations in South Korea a misdemeanor? To be unlawful, does the person must also be producing the noise maliciously? Does the penal code say anything about that?

Why the authors concentrate on multifamily unit dewing’s as opposed to all dwellings? In the U.S. probably most noise complaints occur in single-family homes.

Who is expected to use the smartphones to detect noise, the neighbors mainly, is the Homeowner association or the management company, or is it law enforcement?

I get the engineering and the usefulness of the tool, but my question is about sociology, who use it, and how? Also, why others should care about this research?

The conclusion being one paragraph is hard to read. It’ll be better if the authors divide their thoughts between, the literature shows X and we added X to the conversation. We have 1, 2, 3, recommendations. Futures research could look at X. Overall, this study has made X contribution and we hope it is useful to [indicate audience].

Reviewer 3 Report

Inter-floor noise,  as  the authors have  already mentioned, is one of the leading problems  in urbanized area. It is attributed not only to the Korea, but all the worldwide countries due to growing rate of modern buildings and related issues.

As for the safety, increased noise can badly  influence human health and cause psycho-social stress. It is sometimes hard to  measure  the   noise levels precisely as there are  specific equipment needed. Modern technologies and  devices  are one of great solutions for noise monitoring, whereas precision and quality of  the algorothms of such devices used in smartfphones is a challenge.

There are several recent studies and even reviews dedicated to evaluation of  noise monitoring  systems:

- https://doi.org/10.1155/2019/7634860

-  https://doi.org/10.3390/s19173633

- https://doi.org/10.1007/s40726-018-0090-z

Few studies have also beed  directly dedicated to smart phone applications: 

-https://doi.org/10.1109/UEMCON47517.2019.8993003

- https://doi.org/10.3390/s17040917

This current study shows also an another study on the improvment of such devices based on  smartphone systems sensing inter-floor noise levels!

1) The abstract  briefly  describes purpose of this study and the concept.  However, the  MAE values of 0.2 m minimum error and 3.3. maximum error  do not complain with the information mentioned in the Conclusions part (4.3. determined as the max error.

2)  Did the authors evaluate the accuracy and errors of  the noise measurement (dBA) and vibration velocity?

3) What is the optimal noise measurement range, comparing to  professional sound meters used in indoor  nouse measurements. 

4) The Introduction  describes the problem, the situation in Korea. The study by  Murphy and King testing  noise  measurement apps   have been mentioned. The studies  mentioned above  may also be mentioned during the   literature analysis as it is more dedicated to the problem, but  there is too little evaluation provided on the discussion of the  technical availability of  analogues devices available or tests provided.

5) The concept on the  measurement  provile is  clearly  describet, whereas the   callibration and   evaluation opf precision have not been clearly shown.

6) In Table 2 the  LAm,ax and  LAeq 1 minute description should be corrected.

7) What are the limitations of the measurements in case of big buildings? The  precision of the  measurements should be evaluated as it was mentioned  above.

8) The algorithm for  the evaluation of noise source is clearly written.

9)  The Conclusion part clearly describes results and future applications and future studies

Reviewer 4 Report

Brief Summary

The Technical note presents a procedure aimed at determine the location of a noise source within an inter-floor multi-dwelling house. In the introduction good information about noise disturbances to people and particularly related to dwelling house environment is presented. The bibliography is centered in the case of South Korea where the increase of urban population and density caused an increase of complaints about noise between floors in the last five years. Many of the social conflicts due to noise are due to the impossibility to determine with precision or legal relevance the source of the noise. Monitoring systems do exist but they rely on heavy or complicated equipment that cannot be easily or extensively deployed in houses. The authors propose a smartphone based application that can determine the source of the noise using a grid of 4 smartphones installed within the home. This is a sound magnitude based approach with a Mean Absolute error of 1.4m and allows determining the room from which the noise originates without relying on precise timing of the sound.

Broad Comments

The introduction is adequate for a Technical Note, introduces the extent of the problem, current mitigation techniques and provides enough bibliography.

The room that originates the noise also contains one of the smartphones of the grid; this is not very realistic since in a real scenario the person who originates the disturbance may not be willing to contribute to the monitoring system. How would be the precision or reliability of the method without the sensor closest to the sound source?

Equations (1) to (5) use a rather rigid method in which a regular grid of sensors is required; did you consider the possibility of using an arbitrary sensor placement and then solving a system of equations to determine the sound origin?

Lamelas F. 2019, (Ref [29] in the manuscript) uses sound timing to determine position with 1cm accuracy. No sound timing is used in the present approach in order to determine source position, instead magnitude attenuation is used and this yields an accuracy of around 1.4m, much worse than the accuracy obtained by Lamelas F. but still enough for the purpose of this Multi-Dwelling house application. Can you comment on the reasons for not using sound timing, especially considering that the devices have precise timing from the GPS.

Specific comments (line by line)

Line 37: 1,038 houses? What is this value? Is it the number of apartment houses in South Korea?

71: Figure 1 appears before it is first referenced.

80: Why Ref. to Figure 1 here?

84-85: Here is where Figure 1 should be first referenced.

92: Full stop after [26]

  1. Table 1 not well positioned (it should go after L.119)
  2. Why was this building selected as a Case Study?
  3. Measure

Round 2

Reviewer 3 Report

Authors have made improvments to the manuscript according to the recommendations by the reviewers.

The manuscript is recommended for the further procession and publication in the revised form.